Transcriptomic response of maize primary roots to low temperatures at seedling emergence

Di Fenza Mauro 1 2
Hogg Bridget 1
Grant Jim 3
Barth Susanne susanne.barth@teagasc.ie 2
1 College of Life Sciences, School of Biology and Environmental Sciences, University College Dublin , Dublin , Ireland
2 Crops, Environment & Land Use Programme, Crops Research Centre Oak Park, Teagasc , Carlow , Ireland
3 Research Operations Group, Statistics and Applied Physics Department, Teagasc , Dublin , Ireland
Day David
Electronic publication date: 2017 Jan 5
Publication date: 2017
Volume: 5
Electronic Location ID: e2839
Received 2016 Aug 17; Accepted 2016 Nov 28
Copyright: ©2017 Di Fenza et al.
Copyright year: 2017
Copyright holder: Di Fenza et al.
License: This is an open access article distributed under the terms of the Creative Commons Attribution License, which permits unrestricted use, distribution, reproduction and adaptation in any medium and for any purpose provided that it is properly attributed. For attribution, the original author(s), title, publication source (PeerJ) and either DOI or URL of the article must be cited.
License URL: https://creativecommons.org/licenses/by/4.0/

Keywords: Maize, Cold stress, Chilling tolerance, Transcriptome, Roots, Low temperature

Funding: Irish Department of Agriculture, Food and the Marine (DAFM) Stimulus Research RSF 07 501 This research was supported by the Irish Department of Agriculture, Food and the Marine (DAFM) Stimulus Research grant RSF 07 501. The funders had no role in study design, data collection and analysis, decision to publish, or preparation of the manuscript.

==============================
Background

Maize (Zea mays) is a C4 tropical cereal and its adaptation to temperate climates can be problematic due to low soil temperatures at early stages of establishment.

Methods

In the current study we have firstly investigated the physiological response of twelve maize varieties, from a chilling condition adapted gene pool, to sub-optimal growth temperature during seedling emergence. To identify transcriptomic markers of cold tolerance in already adapted maize genotypes, temperature conditions were set below the optimal growth range in both control and low temperature groups. The conditions were as follows; control (18 °C for 16 h and 12 °C for 8 h) and low temperature (12 °C for 16 h and 6 °C for 8 h). Four genotypes were identified from the condition adapted gene pool with significant contrasting chilling tolerance.

Results

Picker and PR39B29 were the more cold-tolerant lines and Fergus and Codisco were the less cold-tolerant lines. These four varieties were subjected to microarray analysis to identify differentially expressed genes under chilling conditions. Exposure to low temperature during establishment in the maize varieties Picker, PR39B29, Fergus and Codisco, was reflected at the transcriptomic level in the varieties Picker and PR39B29. No significant changes in expression were observed in Fergus and Codisco following chilling stress. A total number of 64 genes were differentially expressed in the two chilling tolerant varieties. These two varieties exhibited contrasting transcriptomic profiles, in which only four genes overlapped.

Discussion

We observed that maize varieties possessing an enhanced root growth ratio under low temperature were more tolerant, which could be an early and inexpensive measure for germplasm screening under controlled conditions. We have identified novel cold inducible genes in an already adapted maize breeding gene pool. This illustrates that further varietal selection for enhanced chilling tolerance is possible in an already preselected gene pool.

Introduction

Worldwide, maize is the foremost cereal product and Europe is the third largest producer of maize grain after the United States and China (www.fao.org). However, along with water availability, low temperatures represent one of the major impediments for plant productivity and geographical distribution in the world (Di Fenza, 2013).

Dent maize is mostly grown in North America and Flint types in Asia, Europe and in Central and South America (Reimer, 2008). Flint corn has a harder kernel compared to dent lines. Besides, flint germplasm has a very low water content, which makes it better suitable to chilling conditions (Revilla et al., 2016). Dent maize in comparison to flint performs better near optimal temperature conditions.

Maize is a C4 tropical plant whose growth range of temperature extends up to 30 °C–35 °C (Presterl et al., 2007; cited in Di Fenza, 2013) and is significantly sensitive to low temperature, particularly in the early growth stages. Despite that hybrids can adapt to lower temperatures than the optimal range (Greaves, 1996), this leads to a steady decline of growth of maize, which stops around 6–8 °C. Prolonged exposure to low temperatures involves irreversible cellular and tissue injury (Greaves, 1996), and the effect is mainly marked in the early growth stages as it impairs several developmental and physiological processes (Marocco, Lorenzoni & Fracheboud, 2005; cited in Di Fenza, 2013).

Chilling stress contributes to yield losses and lower starch and sugar content (Frei, 2000). Chilling affects photosynthesis by an over excitation of Photosystem II (PSII) reaction centres and production of oxygen radicals (ROS). ROS have been shown to produce injurious effects to the photosynthetic apparatus (Nie, Long & Baker, 1992).

In Northern and Central Europe maize is generally sown in the last week of April, when soil temperature is warm enough for seeds to germinate, and harvested in autumn before the first air frost occurs damaging the crop with temperatures below −2 °C. The date of sowing and the date of harvest determine the length of the growing season and therefore the level of maturity and quality of the crop. Early maturing varieties reach maturity earlier; this means that the development of the canopy occurs earlier and so does its closure reducing, this way, the risk of yield losses that can be caused by the first autumn air frost at the end of the growing season. However, despite the improvement in crop quality and yield, these cultivars are still dependent on suitable soil temperatures for the initial establishment of the seedlings and they still benefit from a longer growing season (Di Fenza, 2013).

Maize root growth can occur between 9 °C and 40 °C, but during the early phases of development maize growth is dependent on soil temperature ranging from 10 °C to 17 °C with genotypic variation (Blacklow, 1972). At lower temperatures roots Change. They become swollen behind the tip, get thicker and develop higher number of seminal roots (Farooq et al., 2009). The effect of low temperature on roots may be indirectly reflected on shoot elongation and leaf formation (Hund et al., 2004).

The use of biodegradable polythene films distributed on the soil surface has helped solve the soil temperature issue in practical crop husbandry giving maize growth a significant enhancement (Keane, 2002), but further varietal improvements through breeding are required to make the maize crop more economical in Northern and maritime Europe (Di Fenza, 2013).

A better understanding of the developmental stages that are particularly sensitive to low temperature will help improve maize adaptation to temperate climates. Each physiological and biological process can be more or less susceptible to suboptimal temperatures, depending on what is called thermal threshold, which is the sub optimal temperature at which the maize hybrid is able to maintain high rates of growth. The lower the thermal threshold the higher is the growth rate and the faster is emergence from soil under low temperatures (Greaves, 1996). The thermal threshold is controlled by specific genes which regulate specific processes at specific developmental stages. The combined processes with lower thermal thresholds will result, therefore, in an optimised growth under low temperature conditions.

Plant breeding is still dependant on phenotypic selection, where new hybrids are tested in yield trials and, therefore, selected on the harvestable yield rather than on their ability to cope with chilling temperatures. Chilling tolerance is controlled by genes that are not directly involved in yield, but they contribute to it by conferring tolerance and thus aiding the plant to reach its full yield potential (Greaves, 1996).

The detection of the transcripts and the identification of the genes associated to them will lead, with an appropriate breeding programme, to the transfer of the traits of interest to new hybrids with an improved tolerance to low temperatures (Di Fenza, 2013). Gene expression profiling can be viable with the employment of technologies like microarray and qPCR capable to screen a large set of transcripts or even the entire transcriptome.

In this study, we have investigated the physiological response of primary roots to low temperatures at early seedling emergence in twelve commercially available chilling tolerant maize cultivars differing in kernel type and maturing time. Two pairs of varieties, one with the highest and the other with the lowest growth response were selected for gene expression profiling. As the genotypes were known to be cold tolerant and in order to identify specific genes that are regulated in low temperatures conditions such as in maritime temperate climates, both control and stress climate conditions were set to sub-optimal temperatures such as typical growth conditions in Ireland. This was aimed at the identification of novel transcripts conferring enhanced chilling tolerance in an already adapted maize breeding gene pool.

Material and Methods

Plant material and growth conditions

Untreated maize seeds for the physiological experiment were provided by the seed companies Caussade (France), Pioneer (France) and Codisem (France), for a total of twelve varieties (Table S1). Varieties Algans, Justina and Picker were included in the Irish Recommended List 2008 of the cultivars that have shown a high yield performance under Irish climate conditions in trials. The experimental varieties also differed in the type of kernel (Flint, Dent, and Flint-Dent) and maturity type.

Two independent non seed coating treated 45-seed groups of each variety were germinated in growth chambers (Snijder Microclima 1750, The Netherlands) under control and low temperature conditions on a surface of capillary matting lying over two layers of blotting paper soaked with 100 ml of distilled water. The blotting paper and capillary matting were placed in 52 cm × 42 cm × 9 cm seed trays, which were covered by another inverted seed tray to reduce water loss. The blotting paper was placed below the single layer of capillary matting. Each group of 45 seeds, were arranged as two sub-groups of 21 and 24 seeds in two separate seed trays. Seeds were used directly from the seed bag and placed under two controlled growth conditions. The control temperature regime was set at 18 °C for 16 h and 12 °C for 8 h; the low temperature regime was set at 12 °C for 16 h and 6 °C for 8 h. Control temperature conditions were chosen like at an ideal spring day when maize is being sown in temperate climates like in Ireland. This allowed us to target the identification of transcripts which are up- or down regulated in germplasm which has been bred to perform in the maritime climate of the northern hemisphere. The experiment was conducted in constant dark conditions with 40% of relative humidity.

Germination assessment and growth ratio (GR)

Seed germination was classified as such when the radicle emerged from the meristem and was at least 1 mm long. Shoot measurements were taken as an indirect assessment of root growth performance. Seed germination was recorded in the number of days from sowing to radicle emergence (seed germination data under control and cold stress treatments in Table S1). The length of the primary roots and the shoots were measured on germinated seeds with similar root length under the same growth conditions at 24-hour intervals over a period of five days post-germination (time points). Trays were daily watered with 100ml of distilled water. The response to low temperature was calculated at every time point and was expressed as growth ratio (GR). The GR was calculated dividing the average length of roots and shoots of each variety at low temperature by the average length of root and shoot of each variety at control temperature. GR was expressed in percentage.

Analysis of the physiological response

The experiment was a randomised block design and was conducted as three independent experiments. A three-way ANOVA was fitted including a blocking factor (experiment) and three effects (genotype, treatment and time point) with all the possible interactions of interest (genotype, treatment, time point, genotype × treatment, genotype × time point and genotype × treatment × time point). The measurements over time were made on the same experimental units and a repeated measures analysis incorporating a covariance structure was used to model this lack of independence. P-values < 0.05 were taken as significant. Residual checks were made to ensure that the assumptions of the analysis were met and responses were log transformed to correct skew and/or non-constant variance as appropriate. The analysis was performed with the GenStat statistical software package (VSN International, Hemel Hempstead, Hertfordshire, UK) and the Statistical Analysis System (SAS) software (SAS Institute Inc., Cary, NC, USA).

RNA extraction

Total RNA was isolated of three independent biological replicates from 3 cm maize roots, on days 1, 2, 3, 4, 5 post-germination. Roots were excised and snap frozen in liquid nitrogen and stored at −80 °C. Frozen root samples were homogenised in 1.5 ml microcentrifuge tubes with the use of tube pestles (Sigma-Aldrich, Hamburg, Germany) and used as a starting material for RNA extraction. The isolation of total RNA was carried out with the RNeasy© Plant Mini Kit (Qiagen, Manchester, UK). Samples were treated with DNase I from the RNase-free DNase set (Qiagen). Quantity and quality of RNA were determined with an Agilent 2100 BioAnalyzer (Agilent Technologies, Santa Clara, CA, USA). Only RNA samples with an RNA integrity number (RIN) ≥8 assigned by the BioAnalyzer were used for microarray analysis.

Microarray hybridisation and analysis

Three independent root tissue RNA samples on day 1 post-germination (time point 1) were used for the microarray experiment, for a total of 24 samples. One µg of each RNA sample was hybridised on a 46K 70-mer oligo array developed in the ‘maize oligonucleotide array project’ by the University of Arizona, The Institute for Genomic Research (TIGR) and the University of Wisconsin (Seifert et al., 2012). The 46k array was configured as 4 rows and 12 columns. The intersection of a row with a column represented a subarray. Each subarray consisted in 31 columns and 31 rows. A two-colour microarray was used to compare each variety from the control and the same variety from low temperature according to a loop design. The array hybridisation with Cy3 and Cy5 dyes was conducted by the Institute of Genomic Research in Arizona. RNA samples were sent in RNAstable® microfuge tubes (Biometrica, Las Vegas, NV, USA) according to manufacturer’s instructions. Image acquisition was conducted with a GenePix® scanner (Axon Instruments, Union City, CA, USA) as a service in Arizona.

The analysis of microarray was performed with CARMAweb (Comprehensive R-based Microarray Analysis web service, https://carmaweb.genome.tugraz.at/carma/) a web application based on the R (http://www.r-project.org/) programming language and environment for statistical computing. CARMAweb implements the Bioconductor limma (Linear Model for Microarray Data) package for R, specifically designed for microarray analysis. Data were quality checked, adjusted and normalised before analysis to remove the technical variance and systematic errors without altering the biological variance within the data.

The data were log2 transformed, background corrected and normalised. Background optical noise of the hybridisation was corrected with the minimum method, which consists in giving the half the minimum positive corrected intensities for the array to any intensity that is equal to zero or negative. Within-array normalisation was performed with the print-tip loess method and between-array normalisation was carried out by scaling the expression values.

The statistical analysis was restricted to the 40% of the probe sets in order to reduce the loss of power of the test. When many thousands of simultaneous hypothesis tests are performed, the probability of type I errors grows with the number of tests and the power becomes critical. Therefore, a pre-filtering of the data was applied to remove from consideration the set of genes that is not differentially expressed under any comparison and thus to run the analysis on the 40% of the genes with the biggest variance. Differentially expressed genes were determined by subtracting the average expression of the gene in the control from the average expression of the gene under treatment. Bad array spots were excluded from the analysis. Adjusted p-values were generated using the Benjamini and Hochberg method to correct for multiple testing in the experiment (Benjamini & Hochberg, 1995). The analysis was performed using the limma paired moderated t-test statistics, based on the empirical Bayesian approach.

Microarray data were deposited to GEO (Gene Expression Omnibus) under accession number GSE72508.

Real Time qPCR

Microarray data validation was performed with three independent replicates on day 1 post-germination. Reverse transcription was conducted with 500 ng total RNA, 1 µl 10 mM dNTPs (Bioline, UK), 1 µl oligo-dT(20) primers (Invitrogen, Carlsbad, CA, USA) and incubated at 65 °C for 5 min in a volume of 11 µl. After incubation the solution was chilled on ice for 1 min and 4 µl of 5X FS reaction buffer, 1 µl 0.1M DTT, 1 µl RNase free H2O (Qiagen) and 1 µl SuperScript® III reverse transcriptase (Invitrogen) were added bringing the volume to 20 µl. The reaction was incubated at 50 °C for 60 min and inactivated at 70 °C for 15 min. The solution was brought up to a final volume of 50 µl by adding RNase free H2O.

Relative Real Time PCR was conducted with an ABI 7500/7500 Fast Real Time PCR system (Applied Biosystems, Carlsbad, CA, USA) using the Fast Sybr® Green Master Mix (Applied Biosystems) according to the manufacturer’s instructions, but in a reaction volume of 10 µl.

All primers were designed using the Primer3Plus, an advanced Prime3 designer tool (Untergasser et al., 2007). Lyophilised primers were re-suspended in nuclease free water to a final concentration of 100 pmol µl−1 (mM/L). Re-suspended primers were diluted to the working solution of 10 pmol µl−1. A five series dilution standard curve was used to test primer efficiency. As the efficiency of all the primers was ≥95%, the Livak’s method (ΔΔCt method) was used to calculate the relative expression. Four genes (Table S3) were tested as housekeeping genes with the geNorm algorithm (Vandesompele et al., 2002). Adh and Ef1-α were finally used as housekeeping genes. The target genes used for Real Time qPCR were selected out of the top 100 up- or down regulated genes with the smallest p-values (primer sequences of all candidate genes and housekeeping genes in Table S4).

Three independent replicates on day 1, 2, 3, 4 and 5 post-germination were used to investigate the difference in gene expression of the target genes across five time points in a time series experiment. Fold-change was calculated at every time point by subtracting the average expression of the gene in the control from the average expression of the gene under treatment. The relative expression across time points was calculated by subtracting the fold-change on day 1 post-germination from the fold-change of the other days post-germination.

Results

Physiology and genetics of maize roots grown at low temperatures

The physiological response of maize to low temperature was analysed in twelve maize varieties differing in kernel type and maturity group (Table S1). Overall all of these twelve varieties still had root and shoot growth under chilling treatment and thus all of them could be broadly considered as chilling tolerant (Fig. 1). However for root and shoot growth, there was a significant variety × treatment × time point interaction (Table 1) resulting in an effect of variety on treatment and time point at which the measurements were taken. For this reason the physiological response was calculated as a chilling stress/control ratio (see ‘Materials and Methods’) and not simply the root and shoot growth under chilling stress. The twelve varieties exhibited a root response pattern different to the shoot pattern. However, the shoot growth was measured as in indirect effect of low temperature on root elongation, so, to determine the most cold tolerant and the most cold sensitive genotypes the Tukey’s range test (a multiple comparison method) was performed on the root growth ratio (Fig. 1). The physiological response to low temperatures has led to the identification of two groups of genotypes with contrasting cold tolerance. The varieties Picker and PR39B29 showed the highest resistance to chilling stress in terms of both root and shoot growth, while Algans and Justina presented the lowest degree of tolerance. However, because of their poor ability to germinate under the cold stress temperature regime, Algans and Justina were excluded for subsequent microarray analysis and substituted by the second most sensitive pair of varieties, Codisco and Fergus. Therefore, the four final genotypes used for the gene expression profiling were Picker, PR39B29, Codisco and Fergus.

Figure 1 Growth ratio as response to cold stress.

The growth ratio for roots and shoots was obtained by relating the average length of a root and shoot of cold treatment to the length of root and shoot of control, respectively. Tukey’s range test was carried out to determine the two varieties with the highest combined root and shoot response to cold stress (indicated by *) and lowest combined root and shoot tolerance to cold stress (indicated by **). Note: varieties Algans and Justina were excluded from ranking list due to uneven germination rates (Table S1). Vertical error bars represent mean ± SE (n = 135). The difference between varieties was significant at p < 0.05.

Table 1 Three-way Analysis of variance (ANOVA) tests of genotype, treatment and time point on growth in twelve maize varieties.

Effect	Root P value	Shoot P value	
Replicate	<.0001	<.0001	
Variety	<.0001	<.0001	
Treatment	<.0001	<.0001	
Variety × treatment	<.0001	<.0001	
Time point	<.0001	<.0001	
Variety × time point	<.0001	<.0001	
Treatment × time point	<.0001	<.0001	
Variety × treatment × time point	<.0001	<.0001	

Microarray analysis

The microarray analysis of root tissue showed that the most chilling tolerant varieties Picker and PR39B29 have, each, a set of differentially expressed genes (up and down regulated, p-value < 0.05), while no genes were listed for the other two less chilling tolerant varieties Fergus and Codisco, indicating that no significant change in expression was found in any of the genes analysed (Table 2). The overall number of genes up and down regulated in the two more chilling tolerant varieties amounted to 64 (Fig. S1), which are, however, divided in a group of 39 genes in PR39B29 (Table S5) and 30 genes in Picker (Table S6), as the two varieties exhibited two different transcriptomic patterns in which only four genes were shared, although not all with the same degree of regulation (Tables S5 and S6). These four genes were a RNA binding protein (MZ00003507), a pathogenesis related protein-1—maize (MZ00004486), a hypothetical protein (MZ00022876) and an unknown protein (MZ00041708). The Gene Ontology (GO) functions of the genes were available in the probe dataset. However, no information for MZ00003507 was available and the nearest match obtained for this sequence was an RNA binding protein in Arabidopsis thaliana. Microarray data of our hybridization experiments were deposited to GEO (Gene Expression Omnibus) under accession number GSE72508.

Table 2 List of the nine most significant regulated genes in two cold tolerant maize varieties.

ID and Name are annotation of the NSF Maize Oligonucleotide Array Project. MeanM and MeanA describe the average regulation and the average expression of each gene, resulting from the mean of the values of the biological replicates. False discovery rate (FDR) with the Benjamini & Hochberg’s procedure accounted for the differential gene expression (FDR-adjusted p-value < 0.05).

ID	Variety	Mean M	Mean A	Gene product	Abbreviation	
MZ00003507	Picker	2.43	10.59	RNA binding protein (Arabidopsis thaliana)	RBP	
	PR39B29	−2.78	10.81			
MZ00004486	Picker	−3.66	9.86	Pathogenesis related protein-1—maize (Zea mays)	PR-1	
	PR39B29	−2.42	9.35			
MZ00022876	Picker	2.61	8.66	Hypothetical protein (Oryza sativa—japonica cultivar-group)	Ukw-P (1)	
	PR39B29	2.58	9.60			
MZ00041708	Picker	2.18	10.16	Unknown protein (Oryza sativa—japonica cultivar-group)	Ukw-P (2)	
	PR39B29	2.46	13.30			
MZ00023411	PR39B29	2.46	11.87	22 kDa drought-inducible protein (Saccharum hybrid cultivar)	–	
MZ00026737	PR39B29	2.55	11.62	Peroxidase (Zea mays)	–	
MZ00029223	PR39B29	−2.47	11.38	Putative heat shock protein hsp22 precursor (Oryza sativa—japonica cultivar-group)	–	
MZ00026029	Picker	−2.17	9.82	Probable lipid transfer protein—rice (Oryza sativa)	–	
MZ00037140	Picker	−3.45	11.24	Glucose starvation-induced protein precursor (clone pZSS2)—maize (Zea mays)	–	

qRT-PCR for data validation

Nine differentially expressed genes, including the four in common, were selected to validate the microarray analysis with qRT-PCR. Real Time PCR was performed on three independent biological replicates. The log2 average expression values of the qRT-PCR at time point 1 were correlated to the log2 values (M-values) of the microarray analysis for validation. Spearman’s correlation (rho) for non-parametric distribution was derived from the square root of R2, which was 88%. This strong correlation between the two expression profiling techniques assessed that microarray data were successfully validated by qRT-PCR.

qRT-PCR was also performed to investigate the expression pattern of the genes of interest in Picker and PR39B29 across five time points corresponding to number of days post germination. Gene expression was not maintained over the time (Fig. 2), but was subjected to fluctuation. Except for RBP, the gene expression pattern was similar between the two varieties, in particular for PRP-1.

Figure 2 Gene expression pattern of the significant four genes shared between the two most cold tolerant varieties Picker and PR39B29.

Gene expression patterns were examined over five days post-germination, from day 1 to day 5. Vertical error bars represent mean ± SE (n = 3).

Discussion

Investigation on the cold tolerance in maize has mainly focused on the early phases of growth, as it is known that plant establishment is fundamental for the crop to reach maturity and maximum development (Di Fenza, 2013). Photosynthesis is impacted by low temperatures (Stamp, 1984). To date most research has investigated the effect of sub-optimal temperatures on photosynthesis during leaf development. Sub-optimal growth temperatures lead to impaired chloroplast function through photo inhibition, altered pigment composition and chlorophyll (Greaves, 1996; Marocco, Lorenzoni & Fracheboud, 2005). The type of organ, shoot or root, has a different tolerance to cold which reflect on the photosynthetic performance (Stamp, 1984; Tollenaar & Lee, 2002). When soil temperatures are too low also germination, the heterotrophic phase of development and root development are also impaired.

Often root studies in maize are performed under field conditions which often require destructive sampling techniques. Digital image acquisition is also a possible venue, which can be biased because of the background noise due to soil (Dong et al., 2003) are also used to grow Roots grown under controlled experimental environments in hydroponics (Sanguineti et al., 1998) or sand columns (Ruta et al., 2010) present sampling limitations. Root traits have been measured by recording with a simple camera or photocopier (Liedgens & Richner, 2001; Collins et al., 1987), a scanner (Dong et al., 2003; Hund, Trachsel & Stamp, 2009) and X-ray imaging techniques (Gregory et al., 2003). Advanced software for the quantitative analysis or root growth and the architecture of complex root systems has been developed (Lobet, Pagès & Draye, 2011). Some measurement methods can be invasive, damage the root samples and reduce the sample size. Besides, growth in the field is significantly influenced by several environmental cues, which makes cold tolerance difficult to sunder from other stresses (Riva-Roveda et al., 2016). It is therefore necessary to implement an adequate controlled growth environment, use techniques that are not destructive and take repetitive measurements of the traits of interest, giving temporal information about root growth over a certain period of time. Climate chambers provide appropriate and reproducible conditions to assess cold tolerance and predict the growth potential of maize seedlings by physiological characteristics (Strigens et al., 2013; Stamp, 1986). The growth of maize seedlings on blotting paper for root development analysis used in this work has been outlined previously, with the specific aim of developing a phenotyping platform for non-destructive and repeated measurements of root growth (Hund, Trachsel & Stamp, 2009).

To date, analysis of gene expression in chilling stressed maize roots has not been carried out. Maize genotypes can be distinguished as cold tolerant and cold sensitive varieties: dent hybrids shows a cold sensitive phenotype, whilst flint hybrids are cold tolerant (Revilla et al., 2016). Whereas low temperatures seriously injure the cold sensitive varieties, the cold tolerant genotypes adjust their metabolism to adapt to the environmental conditions. They grow through the activation of metabolic mechanisms that increase the content of specific molecules, such as cryo-protective compounds and antioxidants, but they also involve the down-regulation of some other gene products such as acquaporins (Janská et al., 2010).

Root and shoot growth was carried out in controlled environmental chambers in a range of temperature regimes that are in accordance with previous studies (Blacklow, 1972; Farooq et al., 2009; Marocco, Lorenzoni & Fracheboud, 2005). Germination requires a minimum temperature of approximately 10 °C (Levitt, 1980) and cellular and tissue damage can occur when temperature is below 5 °C (Greaves, 1996). Prolonged exposure below this temperature can seriously injure seedlings that are no longer able to recover (Theocharis, Clément & Barka, 2012). Although, the environmental conditions applied in our study do not cause deleterious effect on the maize seedlings, they both are sub-optimal chilling temperatures, as chilling is defined as a temperature range of 5–15 °C (Nguyen et al., 2009). More specifically, the control temperature applied in this study is defined as a mild-chilling stress compared to the other stress that is defined as strong-chilling (Marocco, Lorenzoni & Fracheboud, 2005). The reason behind this choice is duplex: firstly, we wanted to represent an ideal spring day when maize is being sown in temperate climates like Ireland. This allowed us to target transcripts that are differentially regulated in the germplasm, which has been bred to perform in the maritime climate of the Northern hemisphere; secondly, as the cultivar were cold tolerant, we believed that comparing the gene expression under cold stress with the gene expression under optimal growth conditions would hide the genetic difference among the varieties and would outline a general common response to low temperature. In support of our hypothesis, a recent study has showed that even genotypes with contrasting cold sensitivity regulate thousands of common genes that are probably involved in a general response to cold stress, whilst only a few are responsible for the genetic difference between the germplasms (Sobkowiak et al., 2014). Therefore, by applying more stringent control environmental conditions, we narrowed down our scope to those few genes that could play a dominant role in cold tolerance and that could highlight the genetic difference not only between cold tolerant and cold sensitive cultivars, but also within cold tolerant genotypes. In our study, cold tolerance was assessed phenotypically, on the results of the physiological experiment, which outlined two contrasting groups, of which a pair of varieties each was destined to gene expression profiling. A pair of varities each group allowed us to detect genetic difference between and within the groups and potential different strategies used to cope with cold stress. The microarray analysis outlined a set of differentially expressed genes whose number (64) is consistent with the number of differntially expressed genes in Sobkowiak’s work (66) and five genes (MZ00004711, MZ00018470, MZ00026737, MZ00041500, MZ00043117) are outlined in both studies, although three (MZ00004711, MZ00026737, MZ00041500) were only regulated in Picker and two (MZ00018470, MZ00043117) only in PR39B29. Interestingly, the microarray analysis has revealed that the transcriptome was differently regulated only in two of the four varieties (Picker and PR39B29), precisely, the varieties whose root growth ratio was significantly higher, thus supporting the evidence that the regulation of specific genes confers higher cold tolerance (Theocharis, Clément & Barka, 2012). As previously stated, these results should not be unexpected as the growth temperatures used for this study were sub-optimal, therefore, common cold regulated genes like those controlling the inducer of cbf expression (ICE)-C repeat binding factor/DRE binding factor1 (CBF/DREB1) transcriptional pathway (Miura & Furumoto, 2013) could have been triggered in both control and stress conditions and, therefore, no difference in expression was detected by microarray for those genes. However, transcriptomic differences were still observed in Picker and PR39B29. Codisco and Fergus were probably non able to trigger a specific response to cope with cold stress and therefore, no difference in expression was detected for these varieties.

Although signicant (P < 0.05) contrasting cold tolerance was observed between PR39B29 (flint) and Fergus (dent) as the kernel type would suggest, stastical analysis showed that, in our experiment, neither kernel or maturing time were sufficient conditions for conferring better growth performance (Table S2). Cultivars with the same maturing time or the same kernel exhibited opposite adaptation to low temperatures, or even both cold tolerant and cold sensitive phenotypes (Huski, PR39D60; P < 0.05). However, when showing both mid-early and flint-dent phenotypes (Crazi, Clariti, Algans and Codisco), the cultivars showed no statistical difference among them (P < 0.05) in root growth. In spite of this, Crazi and Clariti also exhibited both cold tolerant and cold sensitive genotypes (P < 0.05), therefore making them unsuitable for assessing cold tolerance. Nevertheless, the number of replicates for each individual trait is too small to assess cold tolerance on a phenotypical basis (Fergus is the only dent variety).

Gene Ontology showed that the differentially expressed genes in Picker and PR39B29 are mostly involved in molecular functions and biological processes, some of which, such as peroxidases, patogenesis related proteins, RNA binding factors and the plant specific NAC transcription factor, have been demonstrated to be induced in response to cold stress (Nie, Long & Baker, 1992; Edreva, 2005; Lorković, 2009; Puranik et al., 2012).

Interestingly, the microarray analysis outlined two different profiles for Picker and PR39B29, suggesting that, even though the two genoytpes had the same (P < 0.05) physiological response to cold stress, they might trigger different biological pathways to cope with low temperatures. This is the case of an increase in pepditases and proteinases that was observed only in PR39B29. Proteolytical enzymes have been shown to be indispensible in maintaining the physiological state of the plant cells by degrading potentially harmful and irreversibly damaged proteins in response to draught stress (Vaseva et al., 2012). Draught stress is strongly associated with cold stress as chilling is responsible for reduced root hydraulic conductance (Aroca et al., 2003). This explains why the NOD protein was also induced. NOD23-like membrane integral proteins belong to a sub-family of root specific acquaporins, which mediate cold acclimation in plants by regulating root hydraulic conductivity (Ahamed et al., 2012). In PR39B29 cold stress also induced the plant-specific NAC protein, which belongs to a major transcription factor family that has been previously demonstrated to be responsible in the adaptation of plants to environmental stress (Puranik et al., 2012).

Picker showed, as opposed to PR39B29, repression of most of the differentially expressed genes, in particular of metabolic enzymes, ribosomal proteins (rpl2, rpl16) and even of an isoform of peroxidases, which are widely known to act against oxidative stress induced by environmental cues (Kocsy et al., 2001). The lpr2 gene has been shown to be down regulated in soybean, where the arrest of protein synthesis represents a strategy that plants use to cope with stress to promote a quick adaptation in stressful environments (Ludwig & Tenhaken, 2001). Maize plants can even enter a stand-by mode to adapt and quickly recover after a mild-chilling stress (Riva-Roveda et al., 2016).

The analysis of the trascriptomes of Picker and PR39B29 has demonstrated that specific genes need to be induced, while others need to be repressed. However, the strategy used by both the cultivars required depletion of proteins and interruption of their synthesis.

Interestingly, four genes (MZ00003507, MZ00004486, MZ00022876 and MZ00041708) were differentially expressed in both Picker and PR39B29. These genes could be representative of a common response used by cold tolerant varieties to cope with low temperatures. The first of these four common genes encodes for the pathogenesis related protein 1 (PR-1). PR-1 belongs to a group of PR-families that are induced in response to several abiotic stressful environmental stimuli, including wounding and low temperatures (Van Loon, Rep & Pieterse, 2006). The second shared gene encodes for a putative RNA binding protein (RBP). RBPs are known to be involved in the post-transcriptional regulation of RNAs, modulating gene expression during development and in the adaptation of plants in response to environmental stresses (Lorković, 2009). However, the expression of this gene is contrasting in the two cold tolerant genotypes: it is induced in Picker as opposed to the majory of the genes being down regulated, and it is repressed in Picker, where most of the genes are up regulated. This contradictory regulation makes its exact role in cold tolerance unclear. The other two genes (MZ000022876 and MZ00041708) encode for hypotethical proteins whose function is unknown and protein family analysis (http://pfam.xfam.org/) did not reveal a match for their protein sequences that could have helped understand their role in cold tolerance. These genes may represent novel cold induced transcripts in an already adapted maize breeding gene pool and should be the basis for more extensive research on transcripts involved in root tissue cold tolerance.

As RPB showed opposite levels of expression between Picker and PR39B29, we have hypothisised that gene regulation might not be maintained over time. The different gene expression profiles between the cold tolerant varieties could not be reflecting two strategies for cold acclimation, but two different stages of a unique strategy. After all, the suspension of the synthetic apparatus previously described in soybean and the growth arrest seen in maize are only transient. Microarray is a useful high troughput screening, but it only captures a photograph of the transcriptome in a specific physiological instant, which is represented by the time point selected. In order to understand whether the difference in gene expression was maintained over time we have investigated the fold change of the four shared genes across five time points. The transcriptomic pattern of these genes showed that the expression was not maintained over time, but it fluctuated. Interestingly, the expression of RBP in PR39B29 went up, while for Picker went down. Even after this analysis the role of RBP remains to be established. As for the other genes, they showed a similar expression pattern between the varieties, suggesting a less variable way to respond to cold stress.

Supplemental Information

Table S1 Maize varieties included in this study

Maize varieties differed in the type of kernel and maturity group. Some varieties were included in the Irish Recommended List 2008 for showing high performance under Irish climate conditions. (*varieties included in the Irish Recommended List 2008 suitable for growing in the open/without plastic; **varieties included in the Irish Recommended List 2008 suitable for growing covered/with plastic; varieties marked in bold were included in the microarray study). Germination rates under control and cold conditions are listed, including standard deviations (in brackets).

Click here for additional data file.

Table S2 Original data and tables of data generated by the Statistical Analysis Software (SAS)

Click here for additional data file.

Table S3 Housekeeping gene information: Name of gene, accession number and bibliographic reference

Click here for additional data file.

Table S4 Primers used for qRT-PCR

Sequences of housekeeping genes and differential expressed candidate genes.

Click here for additional data file.

Table S5 Differential expressed genes in cultivar PR39B29

Click here for additional data file.

Table S6 Differential expressed genes in cultivar Picker

Click here for additional data file.

Figure S7 Venn diagram of the number of differentially expressed and shared genes amongst maize varieties PR39B29 and Picker

Click here for additional data file.

We are thankful to seed companies supplying us with non-coated experimental seed. We are indebted to Thomas F. Gallagher, now retired from University College Dublin, who was the project coordinator of this study and the PhD study supervisor of MF.

Additional Information and Declarations

Competing Interests

Author Contributions

Microarray Data Deposition

Data Availability

The authors declare there are no competing interests.

Mauro Di Fenza performed the experiments, analyzed the data, contributed reagents/materials/analysis tools, wrote the paper, prepared figures and/or tables, reviewed drafts of the paper.

Bridget Hogg performed the experiments, contributed reagents/materials/analysis tools, wrote the paper, reviewed drafts of the paper.

Jim Grant analyzed the data, contributed reagents/materials/analysis tools, wrote the paper, prepared figures and/or tables, reviewed drafts of the paper.

Susanne Barth conceived and designed the experiments, performed the experiments, contributed reagents/materials/analysis tools, wrote the paper, reviewed drafts of the paper.

The following information was supplied regarding the deposition of microarray data:

GEO: GSE72508.

The following information was supplied regarding data availability:

All gene expression data was deposited at GEO. The raw data has been supplied as Supplementary Files with this manuscript.

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
