# Peer review of "Transcriptomic response of maize primary roots to low temperatures at seedling emergence"

_PeerJ, doi:10.7717/peerj.2839_

## Round 0.1 · original submission · Minor Revisions

· Academic Editor

Minor Revisions

It is important that all of the issues raised by the two reviewers are addressed in a revised manuscript.

Reviewer 1 has raised a serious question about experimental design, namely that the control conditions used were actually 'cold' conditions for maize and that this has influenced the outcomes. I agree with this, but realise that you have justified this approach to some degree in the original manuscript. I ask that you spell this out in more detail in the introduction and discussion, and discuss how this influenced the results.

Reviewer 2 has raised a number of more minor issues and makes detailed suggestions on how the manuscript could be improved, especially the Discussion section. A revised manuscript should address all of these.

Reviewer 1 ·

Basic reporting

In this work, the authors study the cold stress mechanisms in primary roots of maize exposed to low temperatuers. The work is important for better adaptation of maize to temperate climate.

Experimental design

What was the reasoning behing the cold conditions used for germination (12C/6C) This is too cold for maize germination. Control conditions used in this work are in reality cold stress conditions. I understand that the authors were interested in learning about the germination response in the spring conditions in Ireland, but the Irish spring conditions are actually the cold conditions for maize. Maize grows optimally 18C or above and that should have been the control conditions while the control conditions used in this work should have been cold. The problem the way I see it is that both the control and cold conditions are stress conditions for maize. This is reflected in the results as the authors did not get many differentially expressed genes. It is not possible now to rectify this but this should be elaborated in the discussion section.

What percentage of seeds were germinated at the control and cold conditions for each variety? Figure 1 shows relative growth ratio, but since some of the varieties failed to germinate this implies that the control conditions were too stressfull. A figure with separate germination rates for control, cold and warmer temperatures (>18C) for each variety would help put this work in the context of global control conditions.

Line 200: “The false discovery rate (FDR) was set at 40% of the genes with the biggest variance across the samples.” What does this mean? FDR threshold should be set as a standard 5% (<0.05). If any other threshold was used it should be clearly mentioned.

Why only four DE genes were common among the two chilling tolerant varieties? The stress mechanisms are known to be well preserved and one would certainly expect a lot more genes to be commonly DE among varieties with similar levels of chilling tolerance. This has to do with the growth conditions and should be elaborated further.

It would be good to also include results from one-way ANOVA for each variety done separately. And simply show a venn diagram of common and uniquely DE genes in different varieties. For example Table 1 results from three way ANOVA does not specify clearly how many DE genes are found at each time-point in each variety.

Table 2 legend states FDR threshold was 0.05% which isn’t stated in Line 200.

Validity of the findings

It is difficult to imagine that the two chilling tolerant maize varieties have so few DE genes. The authors should elaborate on this further in the discussion and conclusion.

Reviewer 2 ·

Basic reporting

Some raw data need to be supplied as supplementary materials, to fulfil the requirements of the journal. This includes:
1) data used to generate Table 1 (featuring statistics from an ANOVA with p <0.001 in every cell!)
2) primer sequences for qPCR, and
3) the qPCR validation of microarray data.

There are also a few grammatical errors that will need to be fixed.

Experimental design

The basis for the investigation is justified, there are interesting questions and the experimental design is adequate.
Good, detailed descriptions of experimental design and statistical analysis.

Validity of the findings

The data are valid and useful, but not exploited to their full potential and more discussion is needed.

Additional comments

The basis for the investigation is justified, there are interesting questions and the experimental design is adequate. There are also some quite interesting findings. The authors identify a pathogenesis-related protein, a hypothetical protein and an unknown protein that were all up-regulated in the more cold-tolerant maize varieties and these warrant further investigation.

Major comments:

More discussion is needed. In particular, more could be done to align this study with other recent reports featuring QTL maps and transcriptomic profiles from various maize lines with differing cold tolerance capabilities. Some of these reports have been cited but either in the introduction, or in vague passing.

Specific comments:

Abstract:
Line 23: remove “and transcriptomic”, as only a physiological analysis was conducted on all twelve varieties.
Line 26-28: inform the reader that Picker and PR39B29 are the more cold-tolerant lines.
Line 28-29: inform the reader that Fergus and Codisco are the less cold-tolerant lines.
Line 30: Wasn’t the total number of regulated genes actually 65, as four of them were common to both Picker and PR39B29?
Lines 29-33: There is a contradiction here. First the authors state that the two chilling-tolerant varieties showed “two different transcriptomic patterns”. Then two sentences later, they claim that the two chilling-tolerant varieties had similar expression patterns, indicating a “common response to chilling”. Did I misread this? If so, please clarify; if not, please correct.
Line 33-35: Was it actually a root:shoot ratio that was calculated? From the methods and Figure 1 it seems that it was root:root (Treated:control) and shoot:shoot (Treated:control) ratios that were calculated.


Introduction:
Line 104: “Recently, a microarray analysis has been conducted to identify cold-regulated genes by chlorophyll fluorescence determination” – this sentence doesn’t make sense. The cited reference investigates transcriptomic profiles of two maize lines with differing levels of cold tolerance. The two lines were defined as cold -sensitive or -tolerant based on differing capabilities for retaining photosynthetic fluorescence properties (e.g. Fv/Fm) under cold treatment.

Materials and Methods:
Good, detailed descriptions of experimental design and statistical analysis.
Primer sequences for qPCR should be made available as a supplementary table.
Line 234-236: Either I am reading it incorrectly, or the calculation of qPCR fold-change across the time series is very strange. The way I understand it, the authors calculated fold changes for control versus cold-treated samples at each timepoint, then subtracted the fold change of day 1 from the fold change of each other day. If interested in how the expression changed during development, irrespective of temperature, then why not just apply Livak’s method using day 1 as a control and each other day as a separate “treatment “, and ignoring the cold-treated samples?

Results:
Line 256: Should read the second most sensitive (not tolerant) pair of varieties.
Figure 1: How were the SE calculated if means were used to calculate the growth ratios? Also, states in the legend that varieties were significantly different (p < 0.05) – but does this refer to just the root growth ratios, or both roots and shoots, and does it mean all cultivars were different to every other cultivar? Or that the more cold-tolerant were significantly different to the more cold-sensitive cultivars? Some of them do not look significantly different to each other, eg. Picker and PR39829 look the same. Need to be more specific, or give an indication as to which varieties are significantly different from which other varieties.
At what time-point were the shoot and root length data collected?
Needs to be a figure or supplemental data for qPCR validation of microarray data.
The raw data for Table 1 need to be made available. Not only is this a requirement of the journal, but also I am unsatisfied with only being able to see a table solely of p<0.001 !

Discussion:
Picker and PR39B29 were characterised as being more cold-tolerant, and expression profiles showed 30 and 39 genes that were up- or down- regulated in some way. Four genes were common to both cultivars, however one of these four (RBP) was up-regulated in one cultivar and down-regulated in the other, so not a common response.
Codisco and Fergus were characterised as being less cold-tolerant based on “growth ratio”, which was clearly more repressed compared to the other two varieties. But there were no significant changes in any gene transcript levels in response to the cold treatment, relative to control plants for these two varieties. This seems very odd; should there not be at least some changes in stress-related transcripts if the plants were indeed more stressed than the controls? Having had a quick look at some of the raw data, I must agree that I found no significant differences between control and treated roots of at least Fergus plants, and perhaps a high level of variability between biological replicates. In addition, as the authors have also stated that the “control” samples were actually grown at sub-optimal temperature and so there may have been an underlying cold response in all four cultivars that was missed because there were no optimal control conditions. In light of this, it would have been good to include an “optimal” control set of samples in this experiment.
In the introduction, the authors provided information on the hardiness of particular types of maize varieties (Flint and Dent) as a result of morphological differences, including kernel structure and water content. This may provide clues as to the types of gene products that may be expected to differ between varieties with different levels of cold tolerance (i.e. genes related to cell structure, metabolism and transport could be influential). It would be good to see some more discussion on this in this section, in light of the findings of the expression profiling, namely decreases in a putative chitinase, a lipid transfer protein and a transmembrane transporter in cold-treated Picker.
The authors also introduced the effects of cold stress on various aspects of morphology and metabolism. For example, metabolic energy content, photosynthesis, ROS, root morphology (swelling and more seminal roots). While there is a very brief mention of some of these aspects in the discussion, I think more can be done here. Picker and PR39B29 both showed a decrease in a peroxidase gene, although the actual isoforms were different, and PR39B29 also showed an increase in an additional peroxidase isoform. Some discussion about how peroxidases might be involved in combatting cold stress would be good.
There was generally no exploration of the genes that were altered in only Picker or only PR39B29, and how they might act in response to cold stress. For example, Picker showed decreases in metabolic enzyme genes (including ribulosebisphosphate carboxylase) and a putative peroxidase gene, as well as increases in several ribosomal protein genes. On the other hand, PR39B29 showed increases in some proteinases/peptidases (as opposed to increased translational apparatus seen in Picker), increases in some transcription factors (NOD and NAC factors, which are interesting in themselves) and an increase in some metabolic enzyme genes (including phosphoenolpyruvate carboxykinase, farnesyl pyrophosphate synthetase and cinnamoyl coA reductase). I think this is important as it may highlight different avenues that different varieties may take to combat cold stress, and might be of interest to the readers.
The authors also introduced other studies that have used QTL mapping and transcriptomics to uncover genes that might be involved in conferring cold tolerance to maize varieties. Would be good for the authors to note which genes were present in QTLs associated with cold tolerance in the previous studies, including Sobkowiak et al. (2014) and Revilla (2016), and discuss in more detail with respect to the types of genes that were expressed differentially between cultivars in the current study. Were there any similartities / differences?
Line 385: “similar but opposite”? Contradiction! Find another way to say this.
Line 401-406: The authors suggest that the measurement of germination and root growth under cold conditions could act as a screen to search for varieties or progeny that are more cold-tolerant. However, we cannot confirm whether this is representative of what happens in the field, because no direct comparisons were made between the filter paper-germinated seedlings and field-, or even pot- grown seedlings. This screen requires validation.
Line 403: The use of the term “growth ratio of root to shoots” is misleading. It implies the calculation of [root]/[shoot], it was actually calculated as [cold root]/[control root] and [cold shoot]/[control shoot], according to the methods section and Figure 1 legend, if I am interpreting these correctly.

---

## Round 0.2 · accepted · Accept

· Academic Editor

Accept

The manuscript has been amended, and additional data added or explained, as requested by the reviewers.